# Vaginal Microbiome in Reproductive Medicine

**DOI:** 10.3390/diagnostics12081948

**Published:** 2022-08-12

**Authors:** Veronika Günther, Leila Allahqoli, Rafal Watrowski, Nicolai Maass, Johannes Ackermann, Sören von Otte, Ibrahim Alkatout

**Affiliations:** 1Department of Obstetrics and Gynecology, University Hospitals Schleswig-Holstein, Campus Kiel, Arnold-Heller-Straße 3 (House C), 24105 Kiel, Germany; 2University Fertility Center, Ambulanzzentrum des UKSH gGmbH, Arnold-Heller-Straße 3 (House C), 24105 Kiel, Germany; 3School of Public Health, Iran University of Medical Sciences (IUMS), Tehran 14535, Iran; 4Faculty of Medicine (Faculty Associate), University of Freiburg, 79104 Freiburg, Germany

**Keywords:** assisted reproductive technologies, microbiome, implantation failure, dysbiosis, pregnancy

## Abstract

The human microbiome has been given increasing importance in recent years. The establishment of sequencing-based technology has made it possible to identify a large number of bacterial species that were previously beyond the scope of culture-based technologies. Just as microbiome diagnostics has emerged as a major point of focus in science, reproductive medicine has developed into a subject of avid interest, particularly with regard to causal research and treatment options for implantation failure. Thus, the vaginal microbiome is discussed as a factor influencing infertility and a promising target for treatment options. The present review provides an overview of current research concerning the impact of the vaginal microbiome on the outcome of reproductive measures. A non-*Lactobacillus*-dominated microbiome was shown to be associated with dysbiosis, possibly even bacterial vaginosis. This imbalance has a negative impact on implantation rates in assisted reproductive technologies and may also be responsible for habitual abortions. Screening of the microbiome in conjunction with antibiotic and/or probiotic treatment appears to be one way of improving pregnancy outcomes.

## 1. Introduction

Assisted reproductive technologies (ARTs) are a common and promising treatment option for couples faced with the problem of infertility. While the success of the technology is influenced by several technical conditions, the true reproductive potential is defined by the quality of the oocyte, spermatozoa, and the maternal environment, which supports the implantation and ongoing development of the embryo. Thus, three unique physiological environments are involved. However, the crucial factors responsible for successful implantation of the embryo are the endometrium, its microbiome, and immunological aspects (Figure 1). As we acquired more knowledge about the human microbiome, it became evident that the microbiome affects the physiologic function of virtually every organ colonized by bacteria [1].

Billions of microbes, such as bacteria, archaea, fungi, phages, and viruses, colonize the human body [2]. The first studies concerning non-pathogenic bacteria colonizing the human body, published in 1885, were those about *Escherichia coli* in the intestine of healthy children [3]. After this time, other commensal bacteria (bacteria that live in harmony with the host) inhabiting different parts of the human body, such as the nasal and oral cavities, the skin, and the urogenital tract, were discovered throughout the twentieth century [3]. Their role was underestimated. In general, microbes were regarded as a threat to human health. In 1988, Whipps et al. defined the term *microbiome* as “a characteristic microbial community occupying a reasonably well-defined habitat which has distinct physio-chemical properties”. Today, this definition is enriched by a dynamic view of microbial activities that result in ecological niches [4]. It is important to distinguish between the terms *microbiota* and *microbiome*: *microbiota* refers to a community of microorganisms in a particular environment, whereas *microbiome* describes a community of microorganisms and their role within a given environment, taking into account environmental conditions and their mutual interaction.

Whereas 80% of the human microbiota reside in the intestinal tract, 9% of the total human microbiota colonize the urogenital tract [5,6]. The urinary tract was previously considered a sterile body niche, but, today, we know that specific bacterial communities in the urinary tract are responsible for a healthy urinary environment. Changes in this microbiome are related to urologic disorders, such as urinary incontinence, urologic cancers, or neurogenic bladder dysfunction [7]. In the vaginal microbiome, *Lactobacilli* dominate the microbial community and are commonly associated with a healthy genitourinary status. *Lactobacilli* produce lactic acid and protect the vagina by maintaining a low pH that is prohibitive to the growth of most bacteria. The uterine cavity used to be regarded as a sterile organ. However, it has its own specific microbiota, which is 100 to 1000 times less dense [8] but dominated by a greater variety of bacterial species and different strains of *Lactobacillus* than the healthy vaginal microbiota [3,9]. Bacteria in the uterus defend the body against infection and play an important role in reproductive outcomes, such as implantation rates and pre-term birth [10].

The vaginal microbiome is of great interest because of its complexity, increasing knowledge of its role in women’s health, and its impact on reproduction [11]. The aim of this review is to analyze the current evidence concerning the relationship between the female genital microbiome and reproductive medicine, its impact, and possible treatment options for dysbiosis, implantation failure, and recurrent pregnancy loss. Despite the existence and importance of the endometrial and urogenital microbiome, this review will focus on the vaginal microbiome.

## 2. Diagnosis of the Microbiome

Data about the microbiome can be obtained by culture-based or sequencing-based technology. Much of the early work describing the human microbiome is derived from culture-based approaches. After a certain period of culturing, a variety of bacterial species can be identified using characteristic cell staining, morphology, or observed biochemical reactions [12]. Culture-based methods are time-consuming and yield information solely about those bacteria whose metabolic substrates are provided by the culture terrain [13]. Thus, many organisms cannot be identified with culture-based techniques, which causes the investigator to underestimate the diversity of the respective microbiome. Culture-based data, while still informative, must be interpreted within the limitations of the respective technology. However, classic culture techniques continue to play a pivotal role in clinical routine. In addition to being more readily available and more affordable—factors not to be underestimated—culture techniques have the critical advantage of revealing more phenotypical traits, which are of interest to clinical decision-making. However, due to numerous limitations inherent in the cultivation process, a large amount of information on different microbial communities in the genital tract, such as their interdependencies, is bound to remain elusive [14].

Molecular methods can be used to detect non-culturable bacteria in the vaginal microbiome [15,16]. Sequencing of the 16S ribosomal RNA (rRNA)-related genes altered the approach to the study of the microbiome [17]. Specifically, 16S rRNA is a component of the 30S small subunit of a prokaryotic ribosome, and the genes coding for this component are used to reconstruct phylogenies due to their slow rate of genomic evolution [17]. Particularly, the hypervariable regions within the gene that serve as molecular fingerprints can be used to identify the genus and species level [18,19]. This sequencing-based technology permits the detection of microorganisms by means of direct extraction and cloning of DNA from a group of organisms in order to obtain further information about the physiology and ecology of the microbiome [1]. The metagenomics represents the whole genome of all microbes in a defined environment, while single-cell genomics refers to the genomes of individual cells that may or may not contain the full genetic repertoire in the microbiota. The success of these culture-independent studies depends primarily on the quality and quantity of metagenomic DNA isolated from the given samples. Therefore, isolation of metagenomic DNA of good quality from a heterogeneous source, such as human feces or the vaginal microbiome, has been a challenging task [20]. Figure 2 shows an overview of some of the most common techniques used to analyze the vaginal microbiome.

## 3. The Vaginal Microbiome in Healthy Women of Reproductive Age

Before addressing in detail the vaginal microbiome and its impact on reproductive outcomes, it is important to understand the microbiome in its physiologic state. The vaginal microbiome plays an important role in protecting the host from various fungal, bacterial, and viral pathogens and forms the first barrier against the external environment in the lower reproductive tract. In cases of vaginal delivery, the vaginal microbiome colonizes the newborn, thus influencing and protecting the infant’s immune system. Depending on the nature and extent of colonization, different microbial populations induce various stimulations in the immune cells of the newborn. Furthermore, exposure to microorganisms at the beginning of life is important for molding the immune system [21]. The vaginal microbiome also plays an important role in the neurodevelopment of the infant. Microbiota can inhibit histone deacetylases (HDAC), affect microglia, produce neurotransmitters, translocate across the blood–brain barrier, and interact with short-chain fatty acids (SCFA) and free fatty acid receptors (FFAR) in the brain. These factors appear to play an important role in the development of the brain [22,23]. Although individual *Lactobacillus* species are present in small numbers in the vagina at all ages, it is the influence of ovarian hormones from menarche to menopause that favor their dominance and importance. The *Lactobacillus* species dominates the lower genital tract in a concentration of 10^7^–10^8^ colony-forming units per gram of vaginal fluid. Under the influence of estrogen, cells of the vagina proliferate substantially and retain glycogen, whereas progesterone promotes cytolysis of these cells. Thus, glycogen is released for *Lactobacilli* to be cleaved into glucose and maltose. This produces several metabolic products, including lactate, and creates an acidic physiological pH [24]. Lactic acid production is a hallmark of a normal vaginal microbiome because the resulting lower vaginal pH (~4.0) creates an unfavorable environment for the growth of pathogenic bacteria [25]. The mean vaginal lactate concentration in vivo is 0.79 ± 0.22%, causing a physiological pH of 3.8 ± 0.2 [26]. Lactic acid blocks histone deacetylases, thus enhancing gene transcription and DNA repair. It induces autophagy in epithelial cells and promotes homeostasis [27]. Bacteria involved in bacterial vaginosis can be inhibited by lactic acid but not by H_2_O_2_ [28].

*Lactobacilli* may be long or short Gram-positive rods but may also be spherical (coccoid) or club-shaped in appearance. They induce cytokines and crosstalk with immune cells of the vaginal epithelium (such as Langerhans cells or dendritic cells), produce lactate, as well as H_2_O_2_, biosurfactants, and coaggregation molecules to protect against adhesion of facultative pathogenic bacteria [27].

The vaginal microbiome was analyzed in various ethnic groups and the studies revealed diverse compositions of microbiota [29,30]. This appears to be important in the clinical context because differences in microbial composition may influence how the respective vaginal environments respond to infection or interact with other imbalances. Apart from ethnicity, the microbiome is also influenced by physiological events, such as the menstrual cycle and pregnancy, and external factors, such as sexual activity, hygiene habits, and medical treatments [31].

Verhelst et al. used a combination of Gram staining and 16S rRNA sequencing to analyze the vaginal microbiome of 197 pregnant women and found differences in the respective dominant *Lactobacillus* species [32]. The following four major species of lactobacilli were identified: *Lactobacillus crispatus* (*L. crispatus*), *Lactobacillus iners* (*L. iners*), *Lactobacillus jensenii* (*L. jensenii*), and *Lactobacillus gasseri* (*L. gasseri*) [16]. Currently, we are aware of 261 *Lactobacillus* species, some of which were given new names in 2020 [33].

The vaginal microbiome may be classified into groups depending on the composition and dominant Lactobacillus species. Ravel et al. established five groups, the so-called community state type (CST), designated as I, II, III, IV, and V, containing 105, 25, 135, 108, and 21 microbial taxa, respectively [25,29]. Grade I was characterized by a dominance of *Lactobacillus crispatus* and was found in 26.2% of the analyzed population, whereas grades II (6.3%), III (34.1%), and V (5.3%) were identified by a dominance of *Lactobacillus gasseri*, *Lactobacillus iners*, and *Lactobacillus jensenii*, respectively [29]. These four main groups were isolated primarily from White and Asian women. Grade IV, found mainly in Black and Hispanic women, was classified as non-*Lactobacillus*-dominated and included primary anaerobic bacteria, such as *Gardnerella*, *Pretovella*, *Corynebacterium*, *Atopobium*, *Megasphaera,* and *Sneathia* [25,29]. A correlation was observed between CST IV and pre-term birth in Caucasian and African women [34,35]. Table 1 summarizes the different CST types.

## 4. Influence of External and Physiological Factors on the Microbiome

Age, pregnancy, sexual activity, smoking, and exogenous hormones are some of the factors that influence the composition of the vaginal microbiome [25].

The vaginal microbiome composition changes across a female’s lifespan from childhood to puberty, during the reproduction phase and pregnancy, as well over the transition period to menopause and after menopause [36]. From the very beginning, the birth mode (Cesarean section or vaginal delivery) seems to be an important event in the initial colonization of the human microbiome and may be associated with long-term health outcomes. Some studies pointed out that women born by Cesarean section had a higher risk for bacterial vaginosis compared to vaginally delivered women [37,38,39]. Due to the influence of placental hormones, for a short while, the newborn has a *Lactobacillus*-dominated vaginal microbiome (*Lactobacillus*, *Prevotella*, and *Snethia* species), similar to that of the mother. Just a handful of studies have addressed the vaginal microbiome of girls during the hormonal resting phase [40]. A diverse microbiome similar to bacterial vaginosis (dominated by Gram-positive and Gram-negative anaerobic bacteria, as well as Gram-negative aerobic bacteria), with a high pH and greater susceptibility to vaginal infection, prevails until the ovaries produce sex hormones at puberty. After this time, the microbiome is gradually dominated by *Lactobacilli* [36].

At the reproductive age, estrogen levels fluctuate during the menstrual cycle: estrogen rises during the follicular phase, reaches its peak at ovulation, and then decreases in the luteal phase. These fluctuations cause changes in the vaginal microbiome, which explains the decrease in *Lactobacillus* species in the genital tract of post-menopausal women due to low estrogen levels [41]. The high levels of circulating estrogens in women of reproductive age cause glycogen deposits in the vaginal epithelium, which favor the growth of glucose-fermenting *Lactobacillus* species [25]. Inter-individual differences in the composition of the vaginal microbiome, as well as changes during a woman`s reproductive lifespan, have been reported [29]. During perimenopause, the vaginal microbiome is similar to premenopausal women but with a shift concerning the *Lactobacillus* concentration, leading from an acidic to a more neutral pH value. Finally, during post-menopause, the vaginal microbiome is dominated by *Gardnerella*, *Prevotella*, and *Atopobium*, accompanied by a very low *lactobacillus* flora, leading to a neutral pH value [36].

Unprotected vaginal intercourse introduces semen into the vagina. Semen is an alkaline substance that temporarily raises vaginal pH and has the potential to bring new microbial species and strains into the vaginal microbiome from the penile microbiota. The microbiotic system is subject to individual shifts daily. Dynamic short-term variations occur in microbiota during the menstrual cycle and through sexual behavior and may remain in a relatively stable balance even with frequent and diverse sexual activity (such as anal intercourse). Masses of *Lactobacillus* species may or may not be detected on certain days, corresponding to the CST, but the system usually returns to its equilibrium [31].

## 5. From Dysbiosis to Bacterial Vaginosis

The reduction in vaginal *Lactobacilli*, increased vaginal pH, and a shift within the *Lactobacilli* population in favor of less H_2_O_2_-producing or less adherent *Lactobacillus* species lead to a dysbalanced vaginal flora and dysbiosis. This does not initially indicate a disease, but the potential of the physiological vaginal content to inhibit the growth of pathogenic germs is reduced, and, consequently, the risk of vaginal infection is increased. Dysbiosis is frequently regarded as a preliminary stage of bacterial vaginosis. In the pathogenesis of vulvovaginal candidosis, a disrupted vaginal flora appears to favor the overgrowth of candida, although it is known that vaginal mycoses are a normal phenomenon in the lactobacillus flora [42]. The prevalence of dysbiosis is unclear, and it usually remains asymptomatic. Even bacterial vaginosis, the most common microbiological disorder of the vaginal environment in adult women, causes characteristic symptoms in just a half of the cases. Figure 3 illustrates the emergence of bacterial vaginosis.

Gardner and Dukes observed a relative deficiency of *Lactobacilli* and an overgrowth of anaerobic bacteria in cases of bacterial vaginosis (BV) and named this vaginal disorder *Haemophilus vaginalis* vaginitis. The authors first described “clue cells” in 1955 [43], whereas Amsel et al. proposed the first diagnostic criteria for BV in 1983 [44]. The lack of H_2_O_2_-producing *Lactobacilli,* as well as an overgrowth of *Gardnerella vaginalis* and other anaerobic Gram-negative rods and Gram-positive cocci, appear to be the most important factors involved in the presence of BV [45]. Further, 16S rRNA Illumina sequencing of isolates from 14 patients with BV showed an immense heterogeneity of species between patients. In other words, no single pathogen is responsible for the emergence of vaginosis [46].

The Amsel criteria are used to establish the diagnosis of vaginosis in clinical practice. Three of the following four criteria should be present: (a) at least 20% clue cells on wet mount microscopy; (b) a ‘fishy’ odor after adding 10% potassium hydroxide to vaginal secretions; (c) vaginal pH greater than 4.5; (d) a thin, homogenous, or milky vaginal discharge [25,44].

Another scoring for BV is the Nugent score through Gram staining, which is based on microscopic visualization of three bacterial morphotypes: a Nugent score of 0–3 is considered normal, 4–6 intermediate microbiota (or dysbiosis), and 7–10 bacterial vaginosis (Table 2) [25,47].

The high recurrence rate of BV is a notable phenomenon. Primary treatment is initially successful in 80–90% of cases, but relapses occur in more than 30% of women within the first three months. Over a period of six years, recurrences occur in more than 50% of women. The bacterial biofilm adhering to the epithelium in chronic relapsing BV cannot be removed with antibiotic therapy and is probably the reason for the high recurrence rate [28].

Several epidemiological studies revealed an association between the composition of the vaginal microbiome and adverse health outcomes. A lower proportion or abundance of *Lactobacillus* and a higher proportion of facultative and obligate anaerobes, such as *Gardnerella*, *Pretovella*, *Atopobium*, or *Sneathia*, appear to be correlated with a higher risk of acquiring sexually transmitted diseases (STDs), including human immunodeficiency virus (HIV), gonorrhea, chlamydia, trichomonas, herpes simplex virus 2 (HSV2), or syphilis, as well as non-sexually transmitted diseases, such as candidiasis or urinary tract infections [48,49,50,51]. Furthermore, vaginosis is linked to the incidence and prevalence of the human papillomavirus (HPV), along with the associated development of cervical intraepithelial neoplasia and a high risk of cervical cancer [52,53].

Bacterial vaginosis affects 20–50% of women of reproductive age. Due to the afore-mentioned higher risk of sexually transmitted diseases, bacterial vaginosis constitutes a risk factor for subfertility and infertility, including tubal infertility after chlamydia or pelvic inflammatory disease (PID). However, bacterial vaginosis without an STD is found in approximately 19% of the infertile population. It remains subclinical quite often and changes the vaginal microbiome in that it shifts from a *Lactobacillus*-species-dominated environment to one dominated by heterogeneous anaerobic bacteria, such as *Gardnerella vaginalis* or *Atopobium vaginae*. Furthermore, a higher risk of pre-term birth and pre-term rupture of fetal membranes has been reported [3,54,55,56,57].

## 6. Vaginal Microbiome and Assisted Reproductive Technologies

The vaginal microbiome has emerged as an important factor in ART during the last few years. Recurrent implantation failure and/or habitual abortion may be a sign of dysbiosis of the vaginal microbiome and is viewed as a starting point for diagnostic investigations and therapy. Nevertheless, established screening methods and valid therapeutic options do not exist because of the novelty of this topic and the small sample sizes investigated in the studies.

Babu et al. conducted a cross-sectional study in India and analyzed 200 vaginal swabs of 84 healthy women (group 1) without any gynecological disorder, and 116 women with fertility problems (group 2) attending fertility clinics [58]. The vaginal flora of group 1 was dominated by *Lactobacillus* species (40; 27.8%), followed by *Micrococcus* (22; 15.3%), *Enterococcus* (16; 11.1%), and *coagulase-negative Staphylococcus* spp. (12; 8.3%). In group 2, on the other hand, the most dominant flora were *Candida* spp. (30; 26.5%) and *Enterococcus* (26; 23%), followed by Gram-negative bacilli, such as *E. coli* (16; 14.1%). The percentage of *Lactobacillus* species was relatively low (4; 3.5%) in women with fertility problems. Asymptomatic vaginosis was present in 32 women (27.6%) of group 2 and six women (7.1%) of group 1 [58]. The authors noted a higher prevalence of asymptomatic vaginosis and more numerous bacteria associated with bacterial vaginosis in women suffering from infertility. Thus, Babu et al. recommend a screening of the vaginal flora, especially in women undergoing infertility treatment [58].

Differences in diagnostic tools, their outcome, and their significance for infertility patients were analyzed by Haahr et al., who compared a quantitative polymerase chain reaction (qPCR) assay with the Nugent scoring system [57]. A cohort of 130 infertile patients, 90% Caucasian, attending two Danish fertility clinics for in vitro fertilization (IVF), were included. Vaginal swabs were obtained from the posterior fornix and analyzed on Gram-stained slides according to Nugent’s criteria. PCR primers were specific for four common *Lactobacillus species, Gardnerella vaginalis*, and *Atopobium vaginae* [57]. The prevalence of bacterial vaginosis according to the Nugent score was 21% (27/130), whereas the prevalence of an abnormal vaginal microbiota was 28% (36/130), defined on qPCR as high concentrations of *Gardnerella vaginalis* and/or *Atopobium vaginae*. The sensitivity and specificity of the qPCR were both 93% in Nugent-defined bacterial vaginosis. Eighty-four patients completed their IVF treatment. The overall clinical pregnancy rate was 35% (29/84). Interestingly, only 9% (2/22) with abnormal vaginal microbiota on qPCR achieved a clinical pregnancy (*p* = 0.004) [57].

In a prospective study from the Netherlands comprising 192 women, vaginal samples were taken before starting the ART procedure [59]. The composition of the vaginal microbiome was determined by the IS-pro technique. The latter is a bacterial profiling technique that detects and amplifies small fragments of bacterial DNA. A prediction model was established in order to anticipate the outcome of fertility treatment. Women with a low percentage of *Lactobacillus* species in their vaginal sample were less likely to have a successful embryo implantation. The prediction model identified a subgroup of women (17.7%, n = 34) who had a low chance of becoming pregnant after fresh embryo transfer [59]. Based on the vaginal microbiota composition, failure was correctly predicted in 32 of 34 women, resulting in a predictive accuracy rate of 94% (26% sensitivity and 97% specificity) [59]. Additionally, the degree of dominance of *Lactobacillus crispatus* was an important factor in predicting pregnancy: the presence of <60% of *L. crispatus* was associated with a high chance of pregnancy because more than a half of these women (50 of 95) became pregnant [59].

A somewhat more extensive microbiological diagnostic study was performed by Selman et al. In a prospective clinical trial comprising 152 patients undergoing IVF treatment, during embryo transfer, the authors collected separate samples for microbial examination from the following sites: the fundus of the vagina, the cervix, the embryo culture medium before and after embryo transfer, the tip of the catheter, and the external sheet. Each sample was cultured separately [60]. Pregnancy rates in patients who tested positive for vaginal–cervical *Entrobacteriaceae* (22.2% versus 51%) and *Staphylococcus* species (17.6% versus 44%) were significantly lower than those in the negative culture group (*p* < 0.001). With regard to other isolated microorganisms, no differences in pregnancy rates were registered [60].

These data suggest that better knowledge of the vaginal microbiome before ART might help to improve pregnancy rates. The heterogeneity of the studies reporting the above-mentioned data hinders their comparison. The studies differed in regard to the time of collection of vaginal samples, hormonal stimulation, fresh or frozen embryo transfer, embryo quality, oocyte donation, and success rates defined as implantation or live birth rates. Moreover, novel and more standardized molecular approaches have improved the ability to detect microorganisms and are expected to yield better results in the future [3].

## 7. Endometrial Microbiome and Assisted Reproductive Technologies

Until a few years ago, the endometrium was considered a sterile organ. However, a large study comprising more than 100 women revealed that the entire sexual tract is colonized by microbes [61]. The microbiota of the vagina/cervix is almost exclusively dominated by *Lactobacilli* and possesses the highest density of bacteria in the entire genital tract. In the endometrium, however, the proportion of *Lactobacilli* decreases and other bacteria, such as *Pseudomonas*, *Acinetobacter*, are added. The proportion of non-*Lactobacilli* continues to increase in the fallopian tubes and the pouch of Douglas. Just a few isolated *Lactobacilli* are found in the latter. Specific microbiological profiles are linked to various anatomical locations of the body and correlate with each other. Clearly, bacterial colonization patterns in a woman interact very strongly with each other; the authors refer to this phenomenon as a microbiota continuum [61].

The conclusions derived by Moreno et al. differ from those mentioned above. The authors analyzed the composition of the endometrium and its outcome after ART [6]. Moreno et al. compared the results of 16S rRNA analyses of paired samples of endometrial fluid and vaginal aspirates in pre-receptive and receptive phases within the menstrual cycle and correlated these with reproductive outcomes. Independent of hormonal status, the comparison with the vaginal microbiota revealed that the endometrial microbiota is not a carryover from the vagina because some bacteria detected in the endometrium were not found in the vagina of the same subject and vice versa [13]. Furthermore, the existing vaginal and endometrial bacterial communities are not identical in every woman. The endometrial microbiota with a non-*Lactobacillus*-dominated flora (<90% of *Lactobacillus* species and >10% of other bacteria) was associated with lower implantation, pregnancy, ongoing pregnancy, and live birth rates. The authors concluded that—on the other hand—a *Lactobacillus*-dominated endometrial microbiome is associated with significantly higher implantation and lower miscarriage rates compared to women with a non-*Lactobacillus*-dominant microbiome. Live birth rates were 58.8% in the *Lactobacillus*-dominated group versus 6.7% in the non-*Lactobacillus*-dominated group [6].

Liu et al. performed an observational study concerning the endometrial microbiome in 130 infertile women with and without chronic endometritis [62]. Plasma cell density based on Syndecan-1 (CD138)-positive cells was measured in endometrium biopsy specimens. Furthermore, the 16SrRNA genes were sequenced as a culture-independent method. Chronic endometritis was diagnosed if the plasma cell density was above the 95th percentile (>5.15 cells per 10 mm^2^) of the reference range in fertile control subjects. In cases of chronic endometritis, the authors found significantly more numerous bacterial taxa, 18 in number, including *Anaerococcus* and *Gardnerella*, and a relative paucity of *Lactobacillus* (especially *L. crispatus*) in the endometrial cavity. In contrast, a relative abundance of *Lactobacillus* was registered in patients without chronic endometritis (1.9% versus 80.7%) [62].

## 8. Vaginal Microbiome and Recurrent Miscarriage

The association between alterations in the vaginal microbiome and implantation failure has been more extensively addressed than the connection between the vaginal microbiome and unexplained recurrent miscarriage. The few studies addressing the subject comprised small sample sizes. Recurrent miscarriage (RM) is defined as ≥3 consecutive idiopathic miscarriages prior to the gestational age of 12 weeks of pregnancy and has been observed in 1–2% of couples [63,64]. According to the American Society for Reproductive Medicine (ASRM), RM may be diagnosed in women after two or more pregnancy losses with clinical (ultrasonography or histopathology) evidence of pregnancy [65]. Fan et al. analyzed the vaginal microbiome using 16S rRNA sequencing of 31 patients with RM and compared these with 27 women who had experienced a normal induced abortion [66]. The authors observed that patients in the RM group frequently had vaginal infections, such as bacterial vaginosis, fungal vaginosis, or mycoplasma infection. All of these may increase the diversity of vaginal flora. As a special proteobacterium, *Pseudomonas* infection was found disproportionately often in cases of RM. *Pseudomonas* is a Gram-negative bacillus, an opportunistic pathogen, and not part of the vaginal flora. *Pseudomonas* infection, which may occur in any part of the body and in any tissue, is common in burns or wounds, in the middle ear, cornea, respiratory tract, and urethra. This bacterium may also cause endocarditis, gastroenteritis, empyema, and even sepsis [66]. The authors conclude that imbalances in the vaginal flora can be transmitted to the uterus through the vagina, where they may activate chemokines to induce a local immune response. This might disrupt the microcirculation of the local immune system and lead to recurrent miscarriage [66].

## 9. Diagnostic Investigation

A vaginal screening of microbiome disbalance may serve as a new method and may significantly improve diagnosis and treatment in patients with RM [66]. Analysis of the vaginal microbiome enables detection of the entire spectrum of germs and corresponding subgroups. In contrast to the former approach, this creates the possibility of targeted antibiotic therapy to eliminate pathogenic germs, and the option of administering probiotics to restore the physiological germ spectrum.

## 10. Therapeutic Approaches

Depending on the spectrum of bacteria found and their subgroups, the clinician may recommend targeted and individualized antibiotic treatment, supplemented if necessary with appropriate probiotics. The risk of recurrent bacterial vaginosis is thus reduced, and couples with “idiopathic infertility” could receive individualized therapy. Antibiotic agents for bacterial vaginosis, such as metronidazole, clindamycin, or dequalinium chloride, have provided short-term improvements [67,68]. Other options to repress the growth of anaerobes associated with bacterial vaginosis and/or support the growth of *Lactobacilli* include estrogen, lactic acid, or boric acid to lower pH [69,70,71]. Some authors recommend the use of probiotics to modulate the vaginal microbiome, either after antibiotic treatment or as primary therapy [72]. A possible outlook into the future represents the transplantation of the vaginal microbiome in patients with refractory bacterial vaginosis [73]. In analogy, fecal microbiota transplantation is the transfer of stool from a healthy donor into the colon of a patient whose disease is a result of an altered microbiome. The goal is to restore the normal microbiota and consequently cure the disease. The most effective and well-studied indication for fecal microbiota transplantation is recurrent clostridium difficile infection [74].

## 11. Antibiotics

Prophylactic antibiotics before embryo transfer might serve as a potential therapeutic approach. These would reduce the level of microbial colonization in the upper genital tract. However, the results of a single randomized controlled trial showed that microbial colonization of the genital tract was reduced with doxycycline but was not associated with a higher clinical pregnancy rate or reduced virulent bacteria [75]. This topic is controversially discussed [57]. In contrast to the first study mentioned above, other groups have shown that the administration of prophylactic antibiotics (ceftriaxone and metronidazole) 48 hours before oocyte retrieval for IVF embryo transfer was associated with significantly higher implantation (*p* < 0.01) and clinical pregnancy rates (*p* = 0.01) [76]. Based on a systematic review, the American Society for Reproductive Medicine strongly advises against the prophylactic use of antibiotics before embryo transfer because of their possible side effects and the chances of developing antibiotic resistance, as well as the lack of evidence in favor of a better pregnancy outcome.

In the event of confirmed anaerobes, the next step is usually antibiotic therapy with metronidazole (oral or vaginal) or clindamycin (oral or vaginal) [77]. One reason for the high rate of re-infection is the different spectrums of activity of metronidazole and clindamycin: *Gardnerella vaginalis* and *Atopobium vaginae* are sensitive to clindamycin, but the simultaneous activity of clindamycin against lactobacilli is disadvantageous. Thus, although *Gardnerella vaginalis* and *Atopobium vaginae* can usually be eliminated by treatment, this also causes a depletion of the protective *Lactobacilli*. Metronidazole—in contrast to clindamycin—is not effective against lactobacilli, while *Gardnerella vaginalis* subgroups A and D are also resistant to metronidazole. Thus, in many cases, metronidazole treatment provides no more than a partial elimination of *Gardnerella vaginalis* [78]. Another reason for frequent recurrences of bacterial vaginosis is the heterogeneous spectrum of germs: in addition to the “typical” pathogens described above, a large number of potentially pathogenic germs may not respond to metronidazole or clindamycin [78]. Table 3 shows treatment options for BV.

## 12. Substances with Lactic Acid

Substances that directly acidify the vaginal milieu create a basis for the rehabilitation of a *Lactobacillus* flora. In addition, an acidic pH has a growth-inhibiting effect on many pathogens. Acidifying preparations, administered intravaginally, may be used in patients with recurrent BV after antibiotic therapy for six to ten days daily, or as “maintenance therapy” once or twice a week for a longer period of time [81].

## 13. Substances with *Lactobacilli* and Probiotics

*Lactobacilli*, given intravaginally or orally as probiotics, were shown to recolonize the vagina [82,83]. Crucial factors in this regard are the strain specificity and individual characteristics of the strains rather than the genus and species of *Lactobacilli* as such. The *Lactobacilli* used for reconstruction of a healthy vaginal flora should be typical strains of the vagina and should have the ability to adhere to sloughed-off vaginal epithelial cells. The latter is an important prerequisite for further growth of the added populations. In addition, *Lactobacilli* should be able to produce H_2_O_2_ and thus prevent recurrent disease [83]. Ideally, they should have antagonistic effects on *Gardnerella vaginalis*, the main germ of BV, and, if possible, also on *Candida albicans* [82]. Clinical studies have confirmed that the administration of *Lactobacilli* orally or vaginally is useful as a means of prophylaxis as well as therapy. To date, we lack a uniform dosing regimen for *Lactobacilli*. However, vaginal application of lactic acid bacteria to restore a healthy vaginal flora is mainly used after therapy or to maintain a normal flora post-menstrually over about three cycles, each for 8–10 days. Sporadic administration does not appear to be useful because, in dysbiotic conditions, it is important to “flood” the vagina with large quantities of *Lactobacilli* (at least 10^8^) repeatedly. Supplementary administration of one or two doses per week may be useful to maintain an achieved status. Modern *Lactobacillus* preparations ideally contain at least two lactic acid bacteria strains typical of the milieu. These should include at least one H_2_O_2_-producing strain (e.g., *L. gasseri*), as well as one additional strain or strains (such *L. rhamnosis*) that exert antagonistic effects on *Gardnerella vaginalis* [82]. Table 4 provides an overview concerning treatment options with lactic acid and substances with Lactobacilli.

Rapid acidification of the vaginal environment is usually sufficient in cases of a shift in vaginal pH as a result of external influences or in cases of slight dysbiosis. In patients with a disrupted or dysbalanced *Lactobacilli* flora, lactic acid bacteria should be used to restore the physiological flora of the vagina permanently and efficiently.

Probiotics targeted for vaginal health are widely available as dietary supplements or vaginal capsules/suppositories. In vaginal applications, probiotics are applied directly at the site of action, whereas orally supplemented probiotics need to first pass through the gastrointestinal tract before migrating to the vaginal tract. Interestingly, both application routes seem to be efficacious [86,87]. Orally administered probiotics, also referred to as food-grade supplements consisting of live microorganisms, may provide additional beneficial effects to vaginal health via the so-called “gut–vagina axis” by balancing gut microbiota and inhibiting/preventing ascension of urogenital pathogens from the rectum to vaginal tract, as well as stimulating the gut and systemic immune system [88,89]. Several demonstrably effective probiotic oral supplements were developed in recent years; the majority of these were used for the treatment of gastrointestinal conditions [90]. The effectiveness of oral probiotics is based on studies that showed a connection between the gastrointestinal and genitourinary lactobacilli colonization; the intestine acts as a kind of reservoir for vaginal lactobacilli colonization [91]. Nevertheless, many oral and vaginal probiotics have been developed to restore a healthy vaginal microbiome and influence reproductive health positively. Additionally, there are several approaches to influence the vaginal microbiome by nutrition: in order to maintain a healthy microbiome, refined sugar, yeasts, and fungus should be avoided, in contrast to leeks, onions, garlic, oats, bananas, and soybeans, which are supposed to maintain a healthy vaginal microbiome [92].

## 14. Conclusions

The above-mentioned data show a correlation between the genital (especially vaginal and endometrial) microbiome and the success of ARTs. A dysbiosis, bacterial vaginosis, or the absence of some *Lactobacilli* species may lead to a failure of implantation. A complex interaction of *Lactobacillus* strains plays an important role in the equilibrium of the healthy vaginal flora, whereas *Lactobacillus crispatus* appears to protect the body against bacterial vaginosis and exert a positive effect on fertility.

In addition, the absence of *Lactobacillus*-dominated flora in the endometrium seems to be linked with recurrent implantation failure. Thus, a vaginal screening of microbiome disbalance may become a new target and may provide effective treatment options. Individualized therapy, including antibiotics and/or probiotics or lactic acids, may alter the vaginal and endometrial microbiome and lead to higher pregnancy rates in ART. Nevertheless, further studies will be needed to obtain more information about standardized study designs and protocols, sampling methods and sizes, and sequencing techniques. Comparable studies will serve as a basis for developing effective therapies.

## Figures and Tables

**Figure 1 diagnostics-12-01948-f001:**
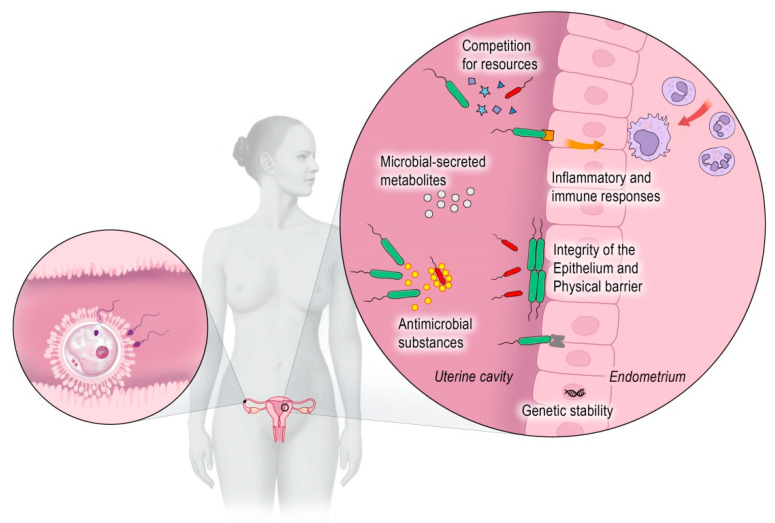
The oocyte, spermatozoa, and the maternal environment are responsible for a successful implantation.

**Figure 2 diagnostics-12-01948-f002:**
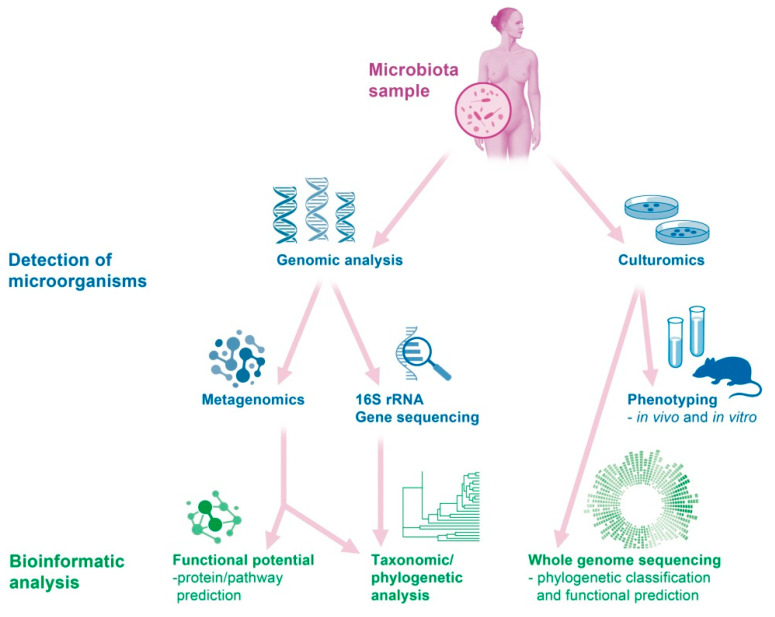
Overview of some of the most common techniques used to analyze the vaginal microbiome.

**Figure 3 diagnostics-12-01948-f003:**
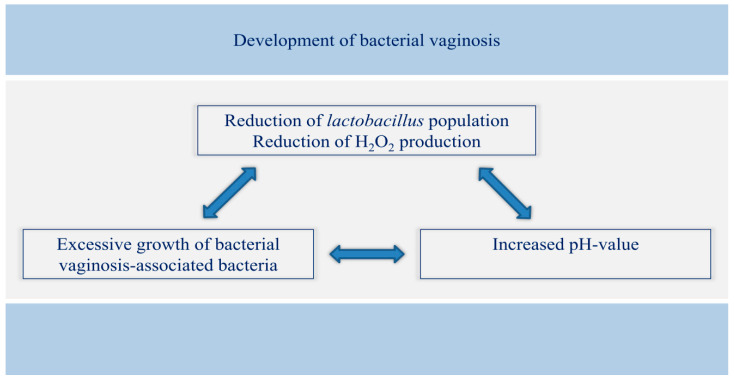
Correlation between *Lactobacillus*, pH value, and pathogenic germs in the emergence of bacterial vaginosis.

**Table 1 diagnostics-12-01948-t001:** Community state type.

Community State Type	Dominating *Lactobacillus* Species	Microbial Taxa	Ethnic Group	pH
**CST I**	*L. crispatus*	105	White and Asian women	4.0 ± 0.3
**CST II**	*L gasseri*	25	Evenly distributed	5.0 ± 0.7
**CST III**	*L. iners*	135	Asian women	4.4 ± 0.6
**CST IV**	Non-*Lactobacillus*-dominated	108	Black and Hispanic women	5.3 ± 0.6
**CST V**	*L. jensenii*	21	White and Asian women	4.7 ± 0.4

CST: community state type, L.: *Lactobacillus* [29].

**Table 2 diagnostics-12-01948-t002:** Nugent scoring of the Gram stain.

	View Multiple Fields (1000× Oil Immersion, High Power)		
Score	Gram-Positive Rods (*Lactobacillus* Species)	Gram-Negative Rods (Such as *Gardnerella* or *Prevotella*)	Curved Gram-Variable Rods (Such as *Mobiluncus curtisii*)
**0**	>30	0	0
**1**	5–30	0–1	1–4
**2**	1–4	1–4	>5
**3**	0–1	5–30	
**4**	0	>30	

Scores 0–3: normal, 4–6: intermediate (vaginal dysbiosis), 7–10: bacterial vaginosis.

**Table 3 diagnostics-12-01948-t003:** Treatment options for bacterial vaginosis [79,80].

Substance	Mode of Administration and Dosage
Metronidazole	2 × 500 mg orally for 7 days alternative: 2 × 2 g orally over 48 h alternative: 2 × 1 g vaginally
Clindamycin	1 × 5 g/day vaginally for 7 days (gel, 2%) alternative: 2–3 × 300 mg orally for 7 days
Dequalinium chloride, vaginal tablets	10 mg vaginally for 6 days

**Table 4 diagnostics-12-01948-t004:** Treatment options with lactic acid and substances with *Lactobacilli* [84,85].

Substance, Mode of Administration	Dosage/Scheme
With lactic acid (vaginal gel or vaginal suppositories)	1 ×/day for 6–10 days after antibiotic therapy alternative: 1–2 ×/week for 6 months as maintenance therapy
With lactobacilli (vaginal tablets/capsules/suppositories), various strains	1 ×/day for 7 days alternative: 1 ×/week for 2 months alternative: postmenstrual 1 ×/day for 8–10 days of the cycle for 3 months
With lactobacilli (oral probiotics), various strains	1–2 ×/day for at least 30 days, better 60 days

## Data Availability

The datasets analyzed for the current study are available from the corresponding author on reasonable request.

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
