# Peer review of "Vaginal Microbiome in Reproductive Medicine"

_diagnostics, 2022, doi:10.3390/diagnostics12081948_

Round 1

Reviewer 1 Report

The review article "Vaginal Microbiome in Reproductive Medicine" by Veronika Günther and colleagues mainly discusses current research concerning the impact of the vaginal microbiome on the outcome of reproductive measures.

Overall, this is a well-written and comprehensive review article describing a very interesting and timely topic. The authors discuss the role of the vaginal microbiome from different angles and describe all technical terms very well, thus both researchers from the microbiome field and the reproductive medicine field can benefit from reading this article.

In principle, I support the publication of the manuscript largely as it is, however, some minor points should be addressed by the authors:

1) References should be added in Table 3, especially since concrete treatment options with recommended dosages are listed.

2) References should be added in Table 4, especially since concrete treatment options with recommended dosages are listed. Furthermore, the authors should mention and refer to Table 4 also in the main text.

Line 111: I think "lower reproductive tract" and not "upper reproductive tract" is meant.

Line 120-22: This sentence should probably be rephrased.

Line 210: I think "Figure 3" and not "Figure 1" is meant.

Lines 354-361: The authors might add how chronic endometritis was defined in this study. If it was defined histopathologically (e.g., via plasma cell density) the authors should include the cutoff values that were used.

Reviewer 2 Report

The Manuscript “
Vaginal Microbiome in Reproductive Medicine” is based upon the use of various pretreatment and treatment methodologies over the vaginal bacteria of traditional as well as modern technologies. Dysbiosis of its harmful aspects and its utilization in women health. It is a really good review, and a lot of effort has been put in it.  Below mentioned comments need to be addressed:

1-     Add about the metagenomics approach

(Paper: An improved methodology to overcome key issues in human fecal metagenomic DNA extraction)

2-     Add about the food resources and probiotics

3-     More details information should be added about the culture method

4-     Authors have not discussed about the bacterial changes due to the different mode of the birth type.

5-     Authors should mention about the fecal transplant method

6-     Authors should add one section about the dysbiosis of the vaginal bacteria in the different age groups.
